# Maternal Mortality Due to Abortion in Brazil: A Temporal, Regional, and Sociodemographic Analysis over the Last Three Decades

**DOI:** 10.3390/healthcare13080951

**Published:** 2025-04-21

**Authors:** Pedro Omar Batista Pereira, Mateus Pinheiro de Souza, Laura Beatriz Argôlo Moreira, Eumar Soares Silva Filho, Edjan da Silva Santos, Amanda Vitória Rodrigues dos Santos, Ana Clara Ferreira Asbeque, Mauro José de Deus Morais, Júlio Eduardo Gomes Pereira, Francisco Naildo Cardoso Leitão

**Affiliations:** Multidisciplinary Laboratory of Studies and Scientific Writing in Health Sciences (LAMEECCS), Federal University of Acre (UFAC), Rio Branco 69920-900, Acre, Brazil; mateus.pinheiro.s@sou.ufac.br (M.P.d.S.); laura.moreira@sou.ufac.br (L.B.A.M.); eumar.filho@sou.ufac.br (E.S.S.F.); edjan.santos@sou.ufac.br (E.d.S.S.); rodrigues.amanda@sou.ufac.br (A.V.R.d.S.); ana.asbeque@sou.ufac.br (A.C.F.A.); mauro.morais@ufac.br (M.J.d.D.M.); julioeduardo.pereira@gmail.com (J.E.G.P.); francisco.leitao@ufac.br (F.N.C.L.)

**Keywords:** women’s health, maternal mortality, abortion, time-series study, Brazil

## Abstract

Background/Objectives: Maternal mortality due to abortion in Brazil has shown a significant decline of 47.37% between 1996 and 2022. This study aims to analyze temporal trends in maternal mortality due to abortion across regions and sociodemographic groups, highlighting disparities and their implications for public health. Methods: Trends were assessed using Prais–Winsten regression models to estimate the annual percentage change (APC). Data were stratified by region and sociodemographic characteristics to identify vulnerable groups. Results: The findings reveal notable regional disparities, with some regions experiencing more pronounced declines than others. Vulnerable sociodemographic groups, including women with lower levels of education and younger age groups, exhibited persistently higher mortality rates. Conclusions: Despite a significant overall reduction in maternal mortality due to abortion, disparities remain among regions and sociodemographic groups. These findings underscore the need for targeted public health policies aimed at reducing inequalities and addressing the needs of the most affected populations.

## 1. Introduction

In Brazil, maternal mortality due to abortion remains a critical public health issue, disproportionately affecting women of childbearing age, particularly those from marginalized groups. This study provides a comprehensive analysis of temporal trends over 27 years, aiming to inform more equitable health policies.

Every two days, a pregnant woman and her baby die due to complications related to abortion in Brazil, which is one of the leading causes of maternal mortality among women of reproductive age (10 to 49 years). Under-reporting, incomplete data, and the illegality of abortion hinder the implementation of effective health policies, making epidemiological studies essential to better understand this issue and guide public health intervention strategies in Brazil.

Globally, maternal mortality remains high, with an estimated 287,000 deaths in 2020, most of which were preventable, especially in low-income countries. Complications from unsafe abortions account for a significant proportion of these deaths, particularly in poorer regions [1]. The situation in Latin America and the Caribbean is also alarming: maternal mortality increased by 15% between 2016 and 2020, reflecting inequality in access to quality health services [2,3].

This global reality is also reflected in Brazil, where regional inequalities and limited access to reproductive healthcare exacerbate the problem. Although previous studies have shown a reduction in maternal mortality rates associated with abortion during different periods [4], this decline is not uniform across all regions of the country [5]. However, the existing literature is limited by short analysis periods, making it difficult to comprehensively evaluate trends over several decades.

In Brazil, young, Black, single women with low educational attainment are the most vulnerable to unsafe abortion, partly due to inadequate access to effective contraceptive methods, lack of sexual education, and social prejudice [4]. The absence of public policies targeting reproductive health exacerbates these conditions, contributing to an increase in unwanted pregnancies and unsafe abortions [6].

The analysis of an unprecedented 27-year time series (1996–2022) provides a comprehensive view of maternal mortality due to abortion in Brazil, considering regional variations and sociodemographic characteristics, and significantly contributes to the international literature on maternal health [7]. In this context, the following research question arises: Have the temporal, regional, and sociodemographic trends of maternal mortality due to abortion in Brazil changed over the last three decades? Additionally, the importance of monitoring severe maternal morbidity and the near miss concept is highlighted, functioning as essential complements for a more accurate analysis of maternal mortality, providing robust data that can inform more effective public policies [8].

Therefore, this study aims to evaluate the temporal and regional trends of maternal mortality due to abortion in Brazil and identify the sociodemographic characteristics of the affected women, contributing to the formulation of more effective and equitable public health policies.

## 2. Materials and Methods

This study is an ecological time-series analysis that examines maternal mortality due to abortion in Brazil. The country has a population of 203,080,756 inhabitants, distributed across 5568 municipalities, according to the Brazilian Institute of Geography and Statistics [9]. Currently, Brazil is divided into five major regions: North, Northeast, Southeast, South, and Central-West. Deaths of women of childbearing age (10–49 years) recorded in the Mortality Information System of the Unified Health System (SIM/SUS) were analyzed, covering all five regions of Brazil, from 1996 to 2022.

The inclusion criteria involved women within this age range whose deaths were classified as abortion-related, according to the Tenth Revision of the International Classification of Diseases (ICD-10). Specific causes included spontaneous abortion (O03), abortion for medical reasons (O04), and other pregnancies ending in abortion (O00–O02, O05–O08). Deaths outside this age range or period, or whose cause was not categorized as abortion, were excluded from the analysis. Complete data for the variables of interest (region, marital status, race/ethnicity, and education) were required for inclusion in the analysis.

The variables analyzed included region (North, Northeast, Southeast, South, and Central-West), marital status (single, married, widowed, and legally separated), race/ethnicity (White, Black, Yellow, Brown, and Indigenous), and education (none, 1 to 3 years, 4 to 7 years, 8 to 11 years, and 12 years or more). The data used in this study were public, extracted from the Mortality Information System (SIM), and made available through the Department of Informatics of the Unified Health System (DATASUS) [10]. Population data were analyzed based on two main databases: one for the years 1996 to 2012, which included census and population count data, and another for the years 2013 to 2022, based on population projections.

To calculate maternal mortality rates related to abortion by marital status and race/ethnicity, population data from 2010 were used, while education data from 2000 were applied. These years were chosen because they are the only available sources on the website of the Brazilian Institute of Geography and Statistics (IBGE) for the respective variables. Although there may be limitations regarding the currency of these data, they were the most complete and reliable for the analysis conducted. For these categories, corrections were made based on tables provided by the IBGE [11]. The projection and rear projection of populations of women aged 10 to 49 were adjusted using an annual growth rate of 0.52%, according to the 2022 census.

Maternal mortality rates due to abortion were calculated using the formula (Number of deaths × 1,000,000)/Population. Data analysis was performed using R software (version 4.4.1), applying the Prais–Winsten generalized linear regression model with robust variance and Durbin–Watson statistics to detect serial autocorrelation. This model, widely used in time-series studies, is particularly suitable for analyzing trends in aggregated data over time, adjusting for autocorrelation in residuals and providing accurate estimates of linear trends [12]. The analyses were conducted using the following R packages: dplyr, ggplot2, openxlsx, prais, readxl, scales, and tidyr. Mean rates, the coefficient of determination (*R*^2^), and the mean annual variation, represented by the coefficient β, were analyzed. *p*-values below 0.05 were considered statistically significant.

The annual percent change (APC) in maternal mortality over the period was calculated using the β coefficient from the Prais–Winsten regression [12]. To determine the total accumulated variation, the Growth Factor (GF) was used. The 95% confidence interval was calculated based on the standard errors of the estimated coefficients, providing a robust and precise analysis of the trends observed over the 27-year study period.

In this study, we used the general abortion-related mortality rate in relation to the number of women of reproductive age (10 to 49 years) as the main measure, instead of the Maternal Mortality Ratio (MMR). The MMR, commonly used in maternal mortality studies, measures the number of maternal deaths per 100,000 live births and is more appropriate for analyses focused on the safety of pregnancy and childbirth. However, our aim is to assess the direct impact of abortion on the mortality of women of reproductive age, and the general abortion-related mortality rate provides a more direct and relevant measure for this purpose.

To ensure the accuracy and reliability of the data, the database was validated by two independent reviewers, who conducted a thorough check for consistency and completeness.

## 3. Results

During the 27-year period, 3648 maternal deaths due to abortion were reported. The overall mortality rate was 576.50 per million women of reproductive age, contributing significantly to the total of 6,327,888 deaths. On average, 135.11 deaths occurred annually (*SD* = 16.10), with the highest number of cases in 2009 (169) and the lowest in 2019 (108). The average rate per million was 2.25 (*SD* = 0.37), with the highest recorded in 1997 (3.12) and the lowest in 2019 (1.68).

The temporal analysis (Figure 1) revealed a significant reduction in maternal mortality rates due to abortion (MMA) in Brazil, with an annual percentage change (APC) of −2.44%, indicating a consistent downward trend (β = −0.011; *p* < 0.001; *R*^2^ = 0.570). The dotted line in the graph represents the predicted values, which are estimates generated by the regression model based on the observed data, projecting the expected trend or behavior over time.

In the regional analysis, a downward trend was observed in all regions of the country. The North region, which had the highest average mortality rates (3.63 deaths per million women/year; 95%CI: 3.29; 3.97), also showed the highest proportions throughout the period, ranging from 21% to 45%, with a sharp increase after 2013. In contrast, the South region registered the largest reduction in APV (−6.82%), going from the highest rate in 1996 to the lowest in 2022 (β = −0.031; *p* < 0.001; *R*^2^ = 0.537), with a significant decrease in proportions from 24% to 5%. The Northeast and Southeast regions showed more stable variations, while the Central-West region experienced more pronounced fluctuations, particularly between 2012 and 2014 (Figure 2).

Table 1 summarizes all observed APCs, highlighting *p*-values and trends. *p*-values indicate the statistical significance of the percentage variations; if below 0.05, they are considered significant, suggesting that the likelihood of the results being due to chance is less than 5%. Trends are classified as follows: increasing (APC greater than 1), decreasing (APC less than −1), and stationary (APC between −1 and 1).

These results highlight the importance of targeted public policies that address regional and sociodemographic inequalities. While some regions and groups have made progress, others, such as Indigenous women and those with 8–11 years of education, face concerning trends. Stagnation in the North and disparities across age groups and racial categories indicate structural barriers. Specific strategies for vulnerable populations, such as education and expanded access to contraceptives, are essential to reducing these disparities.

In the age analysis, the 20–29 years age group recorded the highest maternal mortality rate, with 3.72 cases per million women per year (CI_95%_: 2.32; 5.12). This group also showed the most significant reduction in rates, with an APC of −3.80% (CI_95%_: −5.05; −2.53) and a *p*-value <0.0001. Additionally, other age groups, such as 10–14 and 15–19 years, showed decreasing variations but without statistical significance. For the 15–19 years age group, a significant reduction in maternal mortality rates was observed (β = −0.009; *p* < 0.05; *R*^2^ = 0.156), although this decrease was less pronounced compared to the 20–29 years age group.

Between 1996 and 2022, the age group of 20 to 29 years had the highest proportion of abortion-related deaths, ranging from 28% to 37%, while the extreme age groups (10–14 and 40–49 years) showed significantly lower proportions (Figure 3).

Regarding marital status, the data reveal that the highest maternal mortality rates occurred in the single category, with an average of 2.18 per million women per year (CI_95%_: 1.36; 3.01). In contrast, the analysis showed that widowed and married individuals experienced significant reductions in rates, with an APC of −2.78% (*p* < 0.01) and −2.10% (*p* < 0.05), respectively. Meanwhile, singles and separated individuals demonstrated stationary trends, indicating that despite the high rates, there was no significant change in mortality rates in this population over the period.

From 1996 to 2022, the highest number of abortion-related deaths was observed among singles and separated individuals, with notable peaks in 2010 (54%) and in 2016 (48%), respectively, while widowed individuals consistently showed the lowest proportions, ranging from 0% to 16% (Figure 4).

For race/color, White women recorded a reduction of −2.49% (*p* < 0.001), while Black women showed an increasing trend of 1.97%, without statistical significance (*p >* 0.05). Yellow women experienced a significant reduction of −8.95% (β = −0.041; *p* < 0.05; *R*^2^ = 0.073), marking the largest percentage drop in the studied period. In contrast, indigenous women saw an increase of 13.92% (*p* < 0.01) and had the highest average rate, with 4.21 per million/year (CI_95%_: 2.62; 5.80). Black women also had a significant average rate of 2.47 per million/year (CI_95%_: 1.54; 3.40).

During the period, Black women had the highest proportion of deaths, reaching 40% in 2010, after recording 0% in 1996. In contrast, Yellow women had the lowest proportion, with a single increase to 11% in 2015 and 2022 (Figure 5).

Finally, in the analysis of education, the “none” category showed a significant reduction of −13.51% (β = −0.063; *p* < 0.01; *R*^2^ = 0.417). In contrast, the “8 to 11 years” group experienced a significant increase of 8.61% (*p* < 0.001), and this group also had the highest average rate of cases (*M* = 1.27 per million women/year; CI_95%_: 0.79; 1.76). Furthermore, in the last year analyzed (2022), the deaths recorded in this category accounted for 50.96% of the total cases (Figure 6).

## 4. Discussion

The temporal analysis showed an annual percentage change (APC) of −2.44%, resulting in a cumulative decrease of approximately 47.37% in maternal mortality rates due to abortion (MMA), possibly linked to greater access to prenatal care and reproductive health policies [13]. This reduction, like the 49% decrease in maternal mortality in Brazil between 1990 and 2019 [7], is still not enough, as abortion accounts for 12.5% of maternal deaths, making it the third leading cause [14]. Furthermore, regional variations and socioeconomic inequalities continue to prevent the country from achieving the 75% reduction target proposed by the Millennium Development Goals [15].

These inequalities are more pronounced in low-income regions, where the health infrastructure is poor and access to reproductive services is limited, resulting in higher maternal mortality rates [16]. Unequal public policies and cultural barriers, such as prejudices related to contraception and abortion, hinder access to essential care [17]. Understanding these dynamics is crucial to proposing interventions that promote equity in maternal health access, especially in the most vulnerable regions [15].

Regionally, the trend of reduction in maternal mortality is consistent with previous studies [4,5]. However, the Central-West region showed an increase between 2006 and 2015, suggesting that shorter periods may not reflect long-term trends. The North, in turn, recorded the highest average mortality rates, while the South experienced the greatest reduction in the annual percent change (APC) of −6.82% [4]. This highlights the persistence of regional inequalities and the need for targeted public policies [14,15].

Previous studies corroborate the regional and sociodemographic trends observed in this work. It is reported that inequalities persist, particularly in the North and Northeast regions, reflecting the limited impact of public policies in these areas [4]. Conversely, it is noted that the Southeast and South regions achieved more significant reductions, attributed to improvements in access to reproductive health services and healthcare infrastructure [5]. International research, such as studies conducted in Sub-Saharan Africa, also indicates that regional inequalities and limited access to reproductive services are key determinants of maternal mortality in low- and middle-income countries, aligning with the patterns observed in Brazil [2].

Discrepancies have been identified regarding maternal mortality rates among Indigenous women. These populations are observed to face significantly higher rates compared to other groups, highlighting the need for intercultural health policies [18,19]. Additionally, the illegality of abortion is discussed as a factor that exacerbates under-reporting and compromises the accuracy of maternal mortality data [20], a problem also emphasized in another study that highlighted the limitations of information systems [21].

These findings reinforce the importance of monitoring and evaluating reproductive health policies. Universal access to contraceptive methods and prenatal care is essential to reduce inequalities [22,23]. Thus, the results of this study expand on this evidence, proposing regional interventions targeted at the most vulnerable groups.

In sociodemographic terms, women aged 20 to 29 years had the highest MMA, despite the significant decline in this group (−3.18%) [24]. Complications during pregnancy, childbirth, and the postpartum period reinforce the importance of adequate prenatal care [21]. Regional variations, such as the prevalence of deaths among women over 40 years old in Rio Grande do Sul, indicate the need for specific regional strategies [25].

The largest reduction was among widows (−2.28%). On the other hand, single women had the highest MMA, aligning with the 60.2% of deaths among women without a partner in Recife [26]. Studies in Minas Gerais and São Paulo associate this risk with ineffective contraceptive use [24,27].

The highest rates were among Indigenous and Black women (4.21 and 2.47 per million/year, respectively). There was an increase in hospitalizations of Black, Brown, and Yellow women between 2008 and 2018, reflecting barriers to accessing healthcare [28,29].

The high rate among indigenous women, in line with the MMR in Pará (135.8 per 100,000) [19], and the MMR of 115.14 per 100,000 between 2015 and 2021 [18], reflects reproductive vulnerability. The unacceptably high maternal mortality in this population highlights the need for an intercultural approach to maternal health [30].

The largest percentage reduction occurred among women with no education (−13.51%), suggesting positive effects of educational policies. However, the highest rate was in the “8 to 11 years of education” category (*M* = 1.27 per million/year), which contrasts with studies that associate lower education with higher risks of unsafe abortion, especially in the North, Northeast, and Central-West regions [4,17]. Investing in quality education, particularly in sexual and reproductive health, can help mitigate these risks [22].

The illegality of abortion in Brazil exacerbates under-reporting and hinders effective policies [20]. Criminalization drives vulnerable women to unsafe procedures [6]. Essential measures such as access to obstetric care, prenatal care, reproductive education [23], and continuous family planning can reduce unwanted pregnancies, illegal abortions, and maternal mortality [4].

Although the Mortality Information System (SIM) provides good coverage, the North and Northeast regions still face issues with under-reporting, affecting data accuracy [21]. The COVID-19 pandemic worsened this under-reporting, especially in areas with greater inequality [31]. During the pandemic, the risk of death for pregnant and postpartum women increased by 2.6 times, particularly with the Gamma variant in 2021, highlighting the need for targeted health policies [32].

Under-reporting is a critical limitation of this study, as the quality and availability of data vary across regions of Brazil. In poorer regions, such as the North and Northeast, the under-reporting of abortion cases compromises the accuracy of estimates, underestimating maternal mortality rates [21]. Additionally, the lack of detailed information on the gestational weeks at the time of abortion prevents a deeper analysis of factors associated with maternal mortality at different stages of pregnancy. The database used, although comprehensive, also does not differentiate specific types of abortion, such as chemical, habitual, or induced, which limits the assessment of how these categories might influence mortality outcomes. Lastly, the absence of individual medical histories for patients restricts the analysis of pre-existing factors and clinical conditions that could be associated with maternal mortality.

Despite these limitations, the results of this study provide a comprehensive view of the temporal, regional, and sociodemographic trends in maternal mortality due to abortion in Brazil. Future studies should focus on regional analyses, especially in the North and Northeast, to identify local barriers and solutions that could reduce maternal mortality due to abortion. It is essential to evaluate the impact of public policies, such as access to contraceptives, sexual education, and prenatal care, while also considering socioeconomic and cultural factors, including race/ethnicity and education level. Research should also investigate the effects of health crises, such as COVID-19, on access to obstetric care. Finally, adopting more robust methods, such as sample surveys and independent audits, is crucial to improving data quality and addressing underreporting.

## 5. Conclusions

There has been an overall reduction in maternal mortality due to abortion in Brazil over time, encompassing all regions and most sociodemographic characteristics. However, a detailed analysis of the data revealed that residing in the North region, being aged between 20 and 29 years, being single, being Indigenous, and having 8 to 11 years of education are factors associated with higher rates of maternal mortality due to abortion.

## Figures and Tables

**Figure 1 healthcare-13-00951-f001:**
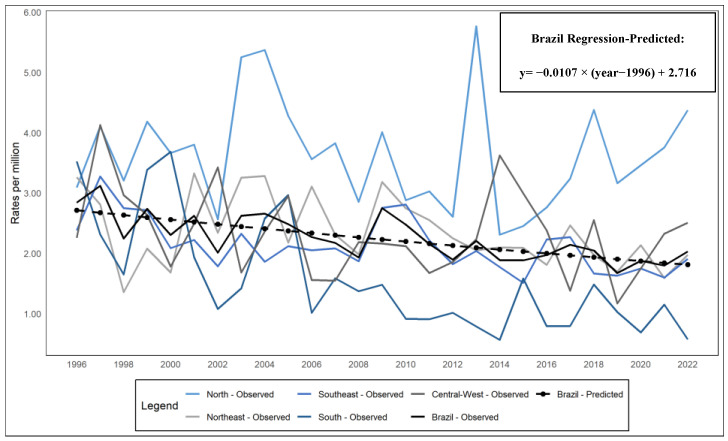
Temporal trend by region, Brazil (1996–2022). Source: DATASUS, 2024.

**Figure 2 healthcare-13-00951-f002:**
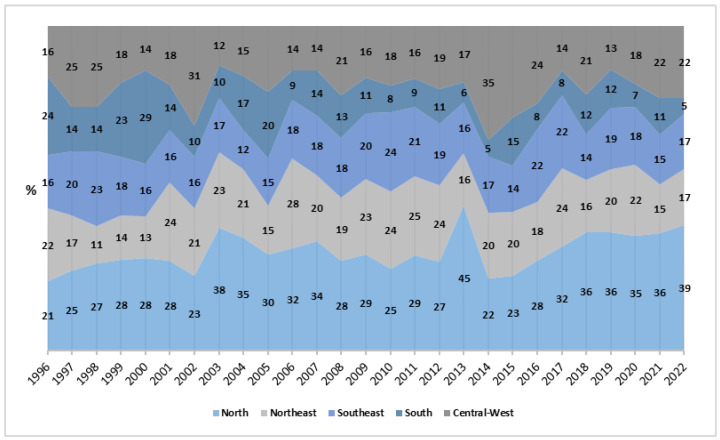
Proportion of rates by region, Brazil (1996–2022). Source: DATASUS, 2024.

**Figure 3 healthcare-13-00951-f003:**
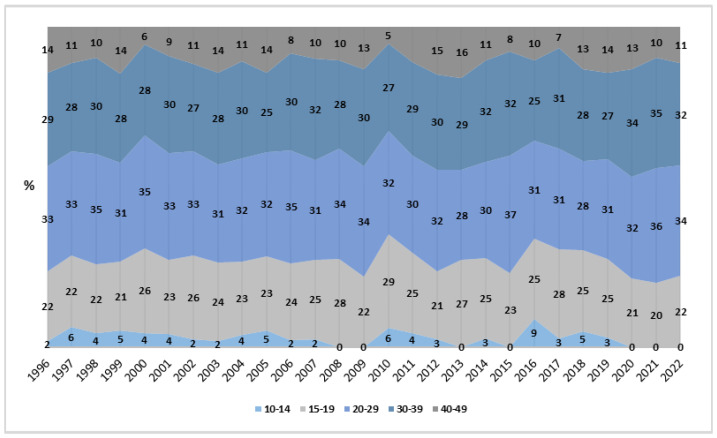
Proportion of rates by age group, Brazil (1996–2022). Source: DATASUS, 2024.

**Figure 4 healthcare-13-00951-f004:**
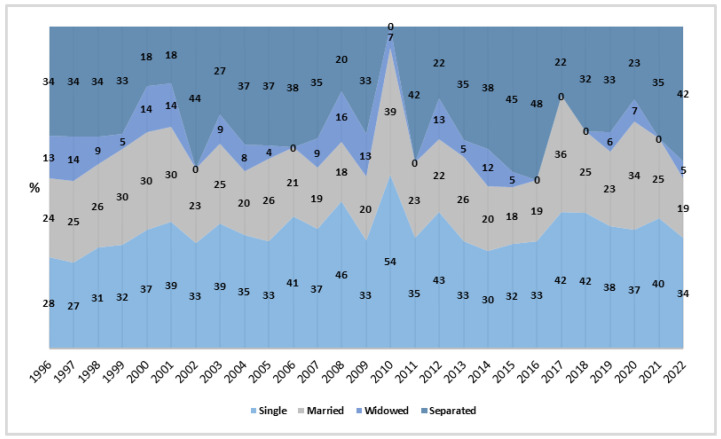
Proportion of rates by marital status, Brazil (1996–2022). Source: DATASUS, 2024.

**Figure 5 healthcare-13-00951-f005:**
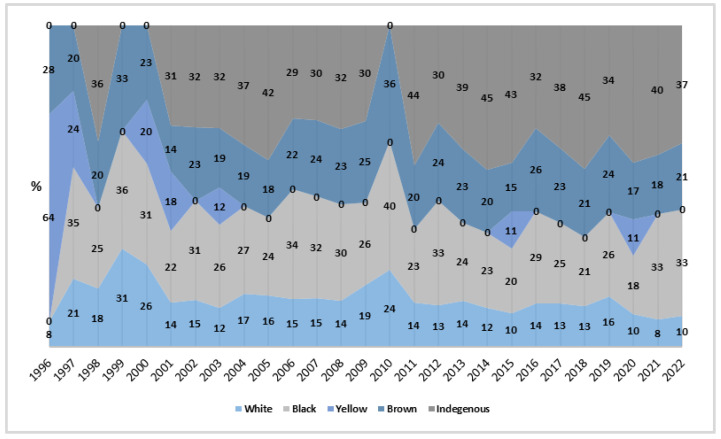
Proportion of rates by race/color, Brazil (1996–2022). Source: DATASUS, 2024.

**Figure 6 healthcare-13-00951-f006:**
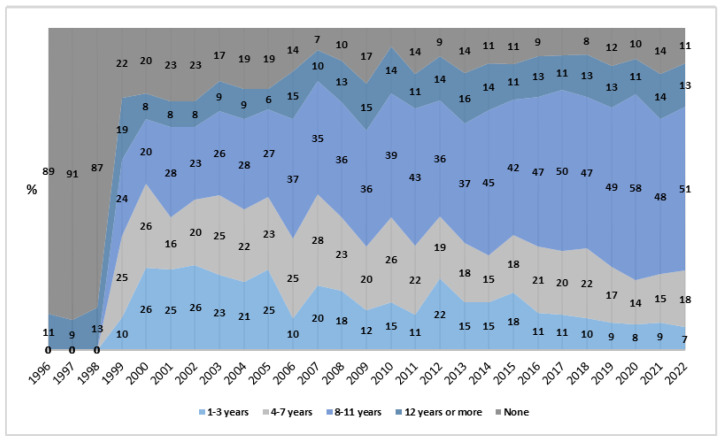
Proportion of rates by education level, Brazil (1996–2022). Source: DATASUS, 2024.

**Table 1 healthcare-13-00951-t001:** Trends in maternal mortality due to abortion, Brazil (1996–2022); source: DATASUS, 2024.

Category	APC (CI_95%_)	*p*-Value	Trend
Regional Brazil	−2.44 (−3.10; −1.77)	<0.0001	Decreasing
North	−0.76 (−2.71; 1.22)	>0.05	Stationary
Northeast	−1.91 (−3.73; −0.05)	>0.05	Decreasing
Southeast	−2.28 (−3.66; −0.88)	<0.005	Decreasing
South	−6.82 (−9.23; −4.35)	<0.0001	Decreasing
Central-West	−1.72 (−4.01; 0.62)	>0.05	Decreasing
Age Brazil	−2.43 (−3.10; −1.76)	<0.0001	Decreasing
10–14	−1.34 (−2.54; −0.11)	>0.05	Decreasing
15–19	−2.02 (−3.69; −0.32)	>0.05	Decreasing
20–29	−3.80 (−5.05; −2.53)	<0.0001	Decreasing
30–39	−1.91 (−2.69; −1.12)	<0.0001	Decreasing
40–49	−1.16 (−2.68; −0.38)	>0.05	Decreasing
Marital Status Brazil	−1.04 (−1.69; −0.39)	<0.01	Decreasing
Single	−0.36 (−1.65; 0.95)	>0.05	Stationary
Married	−2.10 (−3.64; −0.54)	<0.05	Decreasing
Widowed	−2.78 (−4.60; −0.91)	<0.01	Decreasing
Separated	−0.96 (−4.89; 3.12)	>0.05	Stationary
Race Brazil	−0.92 (−1.51; −0.33)	<0.01	Stationary
White	−2.49 (−3.67; −1.29)	<0.001	Decreasing
Black	1.97 (−1.47; 5.53)	>0.05	Increasing
Yellow	−8.95 (−15.75; −1.62)	<0.05	Decreasing
Brown	0.19 (−1.04; 1.44)	>0.05	Stationary
Indigenous	13.92 (6.14; 22.28)	<0.01	Increasing
Education Brazil	−0.88 (−1.42; −0.34)	<0.01	Stationary
1–3 years	−0.15 (−4.48; 4.39)	>0.05	Stationary
4–7 years	1.49 (−2.50; 5.65)	>0.05	Increasing
8–11 years	8.61 (5.34; 11.97)	<0.001	Increasing
12 years or more	0.04 (−0.78; 0.87)	>0.05	Stationary
None	−13.51 (−21.85; −4.29)	<0.01	Decreasing

## Data Availability

The R script and the Excel tables used in this study are available on GitHub at [https://github.com/romah-23/maternal-mortality-analysis], accessed on 20 January 2025.

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
