# Peer review of "Maternal Mortality Due to Abortion in Brazil: A Temporal, Regional, and Sociodemographic Analysis over the Last Three Decades"

_healthcare, 2025, doi:10.3390/healthcare13080951_

Round 1
Reviewer 1 Report
Comments and Suggestions for Authors
This is an interesting manuscript which describe an important health care issue.
In the introduction I would like to understand better the aim of this study. What is the main question? Also please highlight how do this study help in improving health care policy.
After table 1 I would like to understand your opinion related to the trend you have showed in this table.
In discussions I would like to see comparison with several other studies related to this subject.
Thank you.
Author Response
Dear Reviewer 1,
Comment 1:
"This is an interesting manuscript which describes an important health care issue."
Response 1:
Thank you for your positive feedback and for recognizing the relevance of the topic addressed. We are open to considering any additional suggestions that may further improve the manuscript.
-----------------------------------------------------------------------------------------------------------------------
Comment 2:
"In the introduction, I would like to understand better the aim of this study. What is the main question? Also, please highlight how this study helps in improving health care policy."
Response 2:
Thank you for your comment. The main aim was included in lines 59-60: 'Have the temporal, regional, and sociodemographic trends of maternal mortality due to abortion in Brazil changed over the last three decades?'. Additionally, the impact on public policies was highlighted in the introduction and discussion by identifying disparities and proposing targeted interventions.
-----------------------------------------------------------------------------------------------------------------------
Comment 3:
"After table 1, I would like to understand your opinion related to the trend you have shown in this table."
Response 3:
Thank you for your comment. The analysis of trends was detailed between lines 160 and 166, considering the data presented in Table 2. In this section, we discussed persistent regional inequalities in the North and Northeast and the progress in the South, highlighting the influence of public policies and the observed regional disparities.
-----------------------------------------------------------------------------------------------------------------------
Comment 4:
"In discussions, I would like to see comparisons with several other studies related to this subject."
Response 4:
Thank you for your comment. The discussion was expanded to include comparisons with national and international studies between lines 234 and 252. For example, we cited Cardoso et al. (2020), which addresses regional inequalities, and WHO (2023), which emphasizes the importance of intercultural interventions in low-income settings.
-----------------------------------------------------------------------------------------------------------------------
Regarding the aspects to be improved:
(1) Is the research design appropriate?
(2) Are the results clearly presented?
Response:
"Thank you for the comments. The research design was clarified in the methods section, specifically with a detailed explanation of the use of Prais-Winsten regression (lines 104-107), based on the work by Antunes and Cardoso (2015). This approach was chosen due to its suitability for analyzing time series data with autocorrelation of residuals, ensuring greater accuracy in the estimates.
Regarding the presentation of results, the graphs were reformatted to stacked area charts, which enhance the clarity of trends over time and facilitate the interpretation of distributions and total values."
Reviewer 2 Report
Comments and Suggestions for Authors
In this study, the authors evaluated the trends of maternal mortality due to abortion in Brazil. I believe that the study will contribute to the literature in terms of its subject and design. There are also some deficiencies in the manuscript.
1. It has not been investigated whether weeks of abortion have any relationship with maternal mortality.
2. A clear definition of abortion should be made in the material method. Because there are some contradictions in the literature in terms of definition.
3. Is there a relationship between chemical abortions or habitual abortions and maternal mortality?
4. Were only spontaneous abortions included in the study or were there also abortions induced for medical reasons?
5. Why were the medical histories of the patients not investgated in the study?
Author Response
Dear Reviewer 2,
Comment 1:
"It has not been investigated whether weeks of abortion have any relationship with maternal mortality."
Response 1:
Thank you for your comment. We acknowledge that gestational weeks may be a relevant factor in the analysis of maternal mortality due to abortion. However, the data used in this study, derived from the Mortality Information System (SIM), do not include detailed information on gestational weeks. This limitation has been highlighted in the discussion section (lines 289–291), suggesting that future studies should explore this relationship in greater depth.
-----------------------------------------------------------------------------------------------------------------------
Comment 2:
"A clear definition of abortion should be made in the material method. Because there are some contradictions in the literature in terms of definition."
Response 2:
A clear definition of abortion has been included in the methods section, based on ICD-10 codes (O00 to O08), encompassing all forms analyzed: spontaneous abortion (O03), abortion for medical reasons (O04), and other conditions related to abortion (O00, O01, O02, O05-O08). This standardization ensures consistency in the analysis and avoids ambiguities regarding the definition used.
-----------------------------------------------------------------------------------------------------------------------
Comment 3:
"Is there a relationship between chemical abortions or habitual abortions and maternal mortality?"
Response 3:
The database used in this study does not provide details on specific types of abortion, such as chemical or habitual abortions. We recognize the importance of exploring these categories and have discussed in the limitations section (lines 291–294) that future studies with access to more detailed data could investigate this relationship, contributing to a broader understanding of the determinants of maternal mortality.
-----------------------------------------------------------------------------------------------------------------------
Comment 4:
"Were only spontaneous abortions included in the study or were there also abortions induced for medical reasons?"
Response 4:
We clarified in the methods section that all types of abortion recorded under ICD-10 codes were included, encompassing both spontaneous and induced abortions, regardless of their causes (medical or otherwise). This comprehensive approach ensured a broader analysis of maternal mortality associated with abortion in Brazil.
-----------------------------------------------------------------------------------------------------------------------
Comment 5:
"Why were the medical histories of the patients not investigated in the study?"
Response 5:
This study relied on secondary data from the Mortality Information System (SIM), which does not provide detailed medical histories of the patients. This limitation has been included in the discussion section (lines 294–296), highlighting that individual factors, such as pre-existing conditions and clinical history, could not be evaluated. We acknowledge that access to more detailed clinical data would be valuable and suggest that future studies integrate such information for more robust analyses.
-----------------------------------------------------------------------------------------------------------------------
Regarding the aspects to be improved, namely:
(1) Is the research design appropriate?
(2) Are the methods adequately described?
Thank you for the comments. The research design was further detailed in the methods section, including a clearer explanation of the use of Prais-Winsten regression for the analysis of temporal trends, based on the work by Antunes and Cardoso (2015). This approach was chosen due to its suitability for analyzing time series data with autocorrelation of residuals, ensuring greater accuracy in the estimates (lines 104–107).
Regarding the methods, additional details were provided to clarify the inclusion criteria and the variables analyzed. A definition of abortion was included in the methods section, using ICD-10 codes (O00 to O08) to encompass all types of abortion considered in this study: spontaneous (O03), induced for medical reasons (O04), and other related conditions (O00, O01, O02, O05-O08). This standardization avoids ambiguities and ensures consistency in the analysis (lines 116–123). Furthermore, limitations regarding the data sources were addressed in the discussion, acknowledging the lack of detailed information on gestational weeks and medical histories (lines 289–296).
Reviewer 3 Report
Comments and Suggestions for Authors
Dear colleagues, I have read with great attention your work from which your concern and consequently your commitment to improve health care for women and in particular for abortion procedures in Brazil emerges.
The paper, despite the difficulties in recovering all the data on the population, the underestimation determined by the difficulty in recovering data from the registries of causes of maternal death also for the illegality of abortion, presents an analysis of a 27-year time series (1996-2022) that has allowed a complete vision of maternal mortality due to abortion in Brazil, considering regional variations and sociodemographic characteristics, and can contribute significantly to the international literature on maternal health and to the epidemiological studies essential to better understand this problem. It can also guide public health intervention strategies in Brazil. Despite the reduction in the incidence of maternal mortality due to abortion, which in the period considered (27 years) has decreased by 47%, abortion still represents 12.5% ​​of maternal deaths, becoming the third cause of death due to health causes; this means that the social, cultural and health policies implemented in recent years have not been sufficient. Allow me to add and emphasize that the reduction of this tragedy, in addition to counting on a more efficient health service, must above all break down cultural barriers, such as prejudices related to contraception and legal abortion that hinder access to essential care. And the data you presented on improvements that have occurred among the population groups with less education, confirm this. Understanding these dynamics is essential to break down the cultural barriers that push the most vulnerable women to undergo unsafe procedures that are not recommended by official medicine and to propose health interventions in Brazil that program essential measures such as access to obstetric care, prenatal care, reproductive education and family planning. Therefore, I can only congratulate the authors and hope that their work will be a further step towards improving women's health in Brazil.
Author Response
Dear Reviewer 3,
We sincerely thank you for your detailed and encouraging comments on our work. We fully agree with your observations regarding the importance of breaking cultural barriers and implementing more effective public policies to improve women's health in Brazil.
We are particularly grateful for highlighting how the presented data can contribute to the international literature and public health intervention strategies. Our analysis aims precisely to provide a comprehensive view of regional and sociodemographic disparities, as well as to identify existing gaps in reproductive health policies and services.
We hope that this study can indeed serve as a useful tool to support concrete actions that promote universal access to obstetric care, reproductive education, and family planning, reducing inequalities and protecting the most vulnerable women. Thank you very much for your thoughtful remarks and for recognizing the potential impact of our work.
Round 2
Reviewer 2 Report
Comments and Suggestions for Authors
The authors answered clearly my questions. I think that the final version of the manuscript is suitable for publishing in the journal.